# Neonatal outcomes and resuscitation practices following the addition of heart rate-guidance to basic resuscitation

**Jackie K. Patterson**[1]*, **Daniel Ishoso**[2], **Adrien Lokangaka**[2], **Pooja Iyer**[3], **Casey Lowman**[4], **Joar Eilevstjønn**[5], **Ingunn Haug**[5], **Beena D. Kamath-Rayne**[4], **Eric Mafuta**[2], **Helge Myklebust**[5], **Tracy Nolen**[3], **Antoinette Tshefu**[2], **Carl Bose**[1], **Sara Berkelhamer**[6]

1 Department of Pediatrics, University of North Carolina at Chapel Hill, Chapel Hill, NC, United States of America, 2 School of Public Health, University of Kinshasa, Kinshasa, Democratic Republic of Congo, 3 RTI International, Research Triangle Park, NC, United States of America, 4 American Academy of Pediatrics, Itasca, IL, United States of America, 5 Laerdal Medical, Stavanger, Norway, 6 Department of Pediatrics, University of Washington, Seattle, Washington, United States of America

* jackie_patterson@med.unc.edu

**Data Availability Statement:** The data presented in this study are openly available in the USAID Data Development Library at: https://datahub.usaid.gov/

## Abstract

### Aim

To evaluate the impact of heart rate-guided basic resuscitation compared to Helping Babies Breathe on neonatal outcomes and resuscitation practices in the Democratic Republic of the Congo.

### Methods

We conducted a pre-post clinical trial comparing heart rate-guided basic resuscitation to Helping Babies Breathe in three facilities, enrolling in-born neonates ≥28 weeks gestation. We collected observational data during a convenience sample of resuscitations and extracted clinical data from the medical record for all participants. We evaluated our primary outcome of effective breathing at three minutes after birth among newborns not breathing well at 30 seconds after birth employing generalized linear models using maximum likelihood estimation.

### Results

Among 1,284 newborns with observational data, there was no difference in the proportion effectively breathing at three minutes (adjusted relative risk 1.08 [95% CI 0.81, 1.45]). Among 145 receiving bag mask ventilation, time to bag mask ventilation decreased 64.3 seconds during heart rate-guided resuscitation (p<0.001). Among 10,906 enrolled in the trial, perinatal mortality was unchanged (adjusted relative risk 1.19 [95% CI 0.96, 1.48]) and death before discharge increased (adjusted relative risk 1.43 [95% CI 1.03, 1.99]). Expert review of stillborn cases demonstrated a stillbirth misclassification rate of 33.3% during Helping Babies Breathe versus 5.9% in heart rate-guided resuscitation.

Maternal-and-Child-Health/Saving-Lives-at-Birth-DR-Congo-NeoBeat-Study-2017-/keq9-chgt/about_data.

**Funding:** This work was supported by a Saving Lives at Birth Grand Challenge Award, a Thrasher Early Career Award (PI: Jackie Patterson) and a Laerdal Foundation Award (PI: Sara Berkelhamer). The funders had no role in study design, data collection and analysis, decision to publish, or preparation of the manuscript.

**Competing interests:** I have read the journal's policy and the authors of this manuscript have the following competing interests: Jackie K. Patterson is the recipient of a Laerdal Global Health gift and in-kind personnel support for an NIH-funded clinical trial in resuscitation in the DRC. Joar Eilevstjønn, Ingunn Haug and Helge Myklebust are all employed by Laerdal Medical, the company that developed NeoBeat. Beena Kamath-Rayne is an employee of the American Academy of Pediatrics, and was an Associate Editor for the Helping Babies Breathe, 2nd Edition. The other authors have no relevant conflicts of interest to disclose. This does not alter our adherence to PLOS ONE policies on sharing data and materials.

## Conclusion

During heart rate-guided basic resuscitation, time to bag mask ventilation was reduced by greater than one minute. The increase in death before discharge and unchanged perinatal mortality may be due to resuscitation of newborns with a higher risk of mortality who were previously presumed stillborn. A cluster-randomized trial of heart rate-guided basic resuscitation is needed to evaluate its impact on neonatal mortality in low-resource settings.

## Introduction

Ventilation of the lungs is critical to cardiorespiratory transition at birth. Lung aeration promotes vasodilation of the pulmonary vascular bed, ultimately modifying the relationship of the systemic and pulmonary circulatory systems from circuits in parallel to circuits in series [1, 2]. Each year, approximately six million newborns fail to breathe at birth following stimulation. For these infants, early and effective positive pressure ventilation may be life-saving and reduce morbidity [3]; in a study of 459 neonates receiving bag mask ventilation (BMV) in Tanzania, every 30-second delay in BMV increased the risk of death or prolonged admission by 16% [4]. For this reason, evidence-based neonatal resuscitation algorithms consistently recommend initiation of ventilation by 60 seconds after birth for newborns with respiratory depression, followed by careful evaluation of the effectiveness of ventilation.

Newborn heart rate (HR) is a sensitive indicator of the adequacy of spontaneous respiratory effort and thus need for ventilation, as well as the response to resuscitative interventions and effectiveness of BMV. Electrocardiographic (ECG) technology reliably and quickly detects HR, and is recommended by the International Liaison Committee on Resuscitation for assessment of newborn HR during resuscitation [5]. While high-income countries have generally adopted this practice, the relatively high-cost of ECG technology has prevented its adoption in low- and middle-income countries (LMICs) [6, 7].

The neonatal resuscitation algorithm most commonly used in LMICs, Helping Babies Breathe (HBB), relies upon continuous evaluation of breathing as the primary indicator of the need for ventilation [8]. In this algorithm, HR assessment is recommended after one minute of effective BMV. While HBB suggests earlier measurement of HR if a helper is available, the frequent reality of a single provider attending deliveries in LMICs is a significant barrier to earlier HR assessment. Furthermore, HR assessment is typically accomplished with palpation of the umbilical cord or auscultation of the chest, both of which are less accurate and unreliable when compared to ECG [9, 10]. A lack of information about HR during resuscitation of newborns in LMICs may contribute to delayed and ineffective ventilation, impacting clinical outcomes.

Despite an emphasis on assisting infants who do not initiate breathing in the first 60 seconds or "Golden Minute" of life, numerous studies have documented delays in initiation of BMV [11, 12]. Availability of a low-cost, battery-operated HR meter for newborn resuscitation made it possible to evaluate use of ECG-based HR monitoring to guide and improve resuscitation in LMICs [13]. In a clinical trial with midwives in the Democratic Republic of the Congo (DRC), we sought to evaluate the impact of HR-guided basic resuscitation on newborn outcomes and resuscitation practices compared to HBB. Our primary outcome was effective breathing at three minutes after birth. We hypothesized that HR-guided basic resuscitation would increase the proportion of newborns not breathing well at birth who are effectively breathing at three minutes by 50% as compared to resuscitation with HBB alone.

## Materials and methods

### Study design and participants

We conducted a pre-post interventional trial in three urban health facilities in the DRC to evaluate the impact of use of continuous electronic HR monitoring during resuscitation (ClinicalTrials.gov Identifier: NCT03799861) [14]. The study design and participants for this trial have been previously described [15]. The primary objective of this study was to evaluate the impact of resuscitation training in HBB (control group) compared to HR-guided basic resuscitation (intervention group) on effective breathing at three minutes after birth.

Midwives were the primary providers of newborn resuscitation care in the study facilities, and annual births ranged from 1,051 to 4,248 per year [15]. We enrolled all in-born neonates who were ≥ 28 weeks gestation (or birthweight ≥ 1,000 g if gestational age was unavailable).

### Study procedures

At the start of the control phase (Epoch 2a in the primary publication), we conducted standardized resuscitation training for all providers using the HBB 2nd edition curriculum, skills practice and assessments (with facilitation and educational materials in French). We also trained providers to place a battery-operated newborn HR meter called NeoBeat (Laerdal Global Health; Stavanger, Norway) on all non-breathing newborns to aid in distinguishing liveborn from stillborn infants. NeoBeat is a re-usable, C-shaped device that measures HR via dry-electrode technology and is easily placed around the thorax of a newborn by a single provider. Guidance on use of HR to inform resuscitation steps was provided based on the standard HBB curriculum, which emphasizes evaluation of HR after improving ventilation. Following HBB training, providers practiced their skills with a NeoNatalie manikin (Laerdal Global Health; Stavanger, Norway) approximately once monthly. The control period included births which occurred from October 25, 2018 to July 28, 2019.

At the start of the intervention phase (Epoch 2b in the primary publication), [15] we conducted additional training in HR-guided basic resuscitation for all providers. For this curriculum, our study group adapted the HBB action plan and flipchart to include assessment of HR to prompt both initiation and improvement of ventilation. A pilot workshop presenting HR-guided basic resuscitation to HBB-trained providers in Nepal was used to obtain feedback on the developed materials which were then modified for use in the DRC. The final HR-guided action plan included an early assessment of the HR of a non-breathing infant after clearing the airway (if needed), with recommendation for initiation of ventilation with HR<100 beats per minute (bpm) or additional stimulation for newborns with HR≥100 bpm (Fig 1). The cut-off of HR<100 bpm was based off the threshold adopted by the American Academy of Pediatric's Neonatal Resuscitation Program [16]. Following initiation of BMV, the HR-guided action plan included a second assessment of HR in addition to chest rise to determine effectiveness of ventilation, with recommendations to improve ventilation if HR<100 bpm.

Expert facilitators were trained in the HR-guided basic resuscitation curriculum by the study team prior to leading one-day workshops. Consistent with HBB training, the HR-guided basic resuscitation course included a detailed review of the HR-guided algorithm with the support of a flipchart as well as skills practice and assessments. Master trainers also covered content on interpretation of the NeoBeat display to support providers in assessing and using HR throughout the algorithm. The facilitators evaluated participant knowledge and skills using HBB assessment tools including knowledge check and two objective structure clinical exams which were adapted to include HR evaluation to initiate and improve ventilation. Following HR-guided basic resuscitation training, providers continued to practice their skills with the

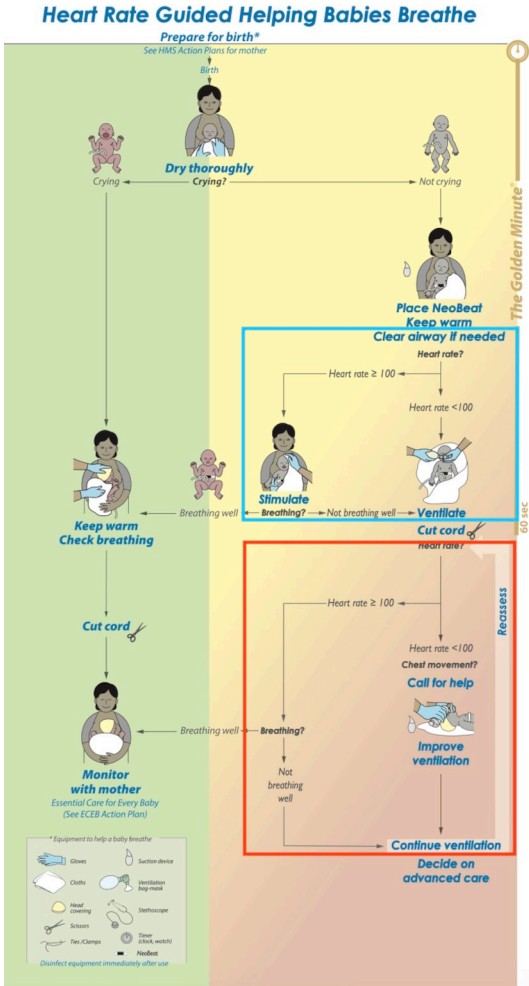

**Fig 1. Heart rate-guided basic resuscitation action plan used during Epoch 2b.** This action plan has two notable differences from the standard HBB action plan. HBB recommends that providers clear the airway of non-crying newborns if needed, followed by stimulation. HR-guided basic resuscitation recommends evaluation of HR after clearing the airway, with progression to ventilation without additional stimulation if HR<100 beats per minute (bpm) (see blue box). HBB predominantly relies on assessment of chest rise to determine the effectiveness of ventilation and need for improvement of ventilation. HR-guided basic resuscitation recommends evaluation of HR after initiating ventilation, with progression to improving ventilation if HR<100bpm. Reassessments result in repeated evaluation of HR throughout the provision of ventilation (see red box). Reprinted from American Academy of Pediatrics Helping Babies Breath Action Plan [8] under a CC BY license, with permission from the American Academy of Pediatrics, original copyright 2016.

NeoNatalie manikin using the HR-guided OSCEs approximately once monthly. The intervention period included births occurring from 8/17/19 to 6/22/20.

As described in the primary manuscript, all demographic and outcome data (liveborn vs stillborn, death before discharge) were collected via medical record abstraction and reflect the outcome as determined by the provider [15]. Research staff with a clinical background (nurse or physician) observed a convenience sample of births and recorded events on the Liveborn Observation App which was synchronized with data collected via NeoBeat [17]. These staff documented start and stop times of key resuscitation actions (skin to skin, drying/stimulation, suction, BMV) and the time of cord clamping and initiation of steps to improve BMV. Research staff continuously evaluated the breathing status of the newborn and recorded this

status in the Liveborn Observation App as not breathing, gasping/slow breathing or breathing well. Research staff underwent rigorous training to ensure accurate observational data collection, and each obtained competency in documenting resuscitation care prior to the initiation of observations in the trial per the procedures previously described [18]. HR data were analyzed for all resuscitations that both applied NeoBeat and were observed with the LIVEBORN app.

## Outcomes and statistical analysis

Our primary outcome for this study was effective breathing at three minutes after birth among newborns who were not breathing well at 30 seconds after birth. We defined effective breathing based on the HBB definition of assessment for 'breathing well,' i.e., crying OR breathing quietly and regularly. We anticipated that HR-guided basic resuscitation would prompt earlier initiation of BMV as well as more efficient achievement of effective BMV. Based on HBB implementation studies that indicate BMV is infrequently administered within the Golden Minute even after training [19, 20], we expected that the vast majority of non-breathing newborns who were then breathing by one minute responded to stimulation [21]. To capture the population of newborns who require BMV to initiate spontaneous breathing (i.e., those in secondary apnea), we defined the timepoint for evaluating the relative impact of HR-guided basic resuscitation on effective breathing as three minutes.

Our secondary outcomes included time to effective breathing, HR≥100 bpm at three minutes after birth, time to HR≥100 bpm, one and five minute APGARs, death before discharge and perinatal mortality (defined as total stillbirths + death before discharge). While we did not anticipate HR-guided basic resuscitation to change the outcome of antepartum stillbirths, we elected to include all stillbirths in this definition of perinatal mortality due to concern for the inaccuracy of clinical exam in distinguishing the timing of stillbirth; this decision is also in keeping with the primary trial in which this substudy was conducted. We chose HR≥100 bpm at three minutes after birth based on prior literature demonstrating the association between this indicator and improved survival among newborns requiring resuscitation [22]. We also evaluated key provider actions as secondary outcomes including initiation time and duration of BMV, drying/stimulation and suctioning.

In post-hoc analysis, we conducted expert review of all stillborn cases for which we had HR data to determine the rate of misclassification based on presence of a documented HR as previously described [15].

We used descriptive statistics to evaluate scores on HR-guided basic resuscitation assessments. For both primary and secondary outcomes, we employed generalized linear models using maximum likelihood estimation. The models all used a canonical logit link function and controlled for epoch (HBB vs HR-guided groups), facility, gestational age and maternal age. All continuous outcomes were evaluated using linear regression and all categorical outcomes were dichotomized and evaluated using logistic regression. While there were observed differences in delivery mode, we find that this is due to the small sample size of cesarean deliveries and not an actual difference; as such, we did not control for mode of delivery in our models.

## Ethics approval

This trial was approved by both the University of North Carolina Institutional Review Board (IRB) (#17–1688) and the DRC National Ethics Committee (#058/CNES/BN/PMMF/2017). These IRBs granted a waiver of informed consent for newborns participating in the trial from facilities B and C. Due to the implementation of a sub-study at facility A which involved HR monitoring of all newborns, we obtained written informed consent for participation in the

**Table 1. Provider knowledge and skill with HR-guided basic resuscitation training.**

| Assessment<br>Median (IQR) | Provider scores<br>N = 56 |
| --- | --- |
| Pre-Knowledge Check | 80.0 (70.0, 92.5) |
| Post-Knowledge Check | 95.0 (90.0, 100.0) |
| Post OSCE A | 92.3 (84.6, 100.0) |
| Post OSCE B | 81.8 (77.3, 86.4) |

trial at facility A. If we were unable to obtain consent prior to birth, we obtained maternal consent for the use of observational data following birth.

## Results

Fifty-nine providers were trained in HBB for the control period and fifty-six were trained in HR-guided basic resuscitation. As previously published, providers achieved a median knowledge score of 79% and a median post OSCE A score of 79% following HBB training [18]. Providers trained in HR-guided basic resuscitation scored a median of 80 on the pre-knowledge check compared to a median of 95 on the post-knowledge check (Table 1). The median score for OSCE A was 92.3% and for OSCE B 81.8%, both above the passing score of 77%.

Data were collected from 6,185 livebirths in the control period and 4,721 livebirths during the intervention (Fig 2). Maternal age, gestational age and mode of delivery were statistically significantly different during HBB compared to HR-guided basic resuscitation (Table 2).

There was no difference in 1-minute or 5-minute APGAR scores ≤5 comparing HR-guided basic resuscitation to HBB (Table 3). Death before discharge increased with an adjusted relative risk of 1.59 (95% CI 1.13, 2.23; Table 3). The adjusted relative risk of perinatal mortality was 1.19 (95% CI 0.96, 1.48) when adjusted for facility only. For the subset of cases documented as stillborn in the medical record which also had accompanying observational and HR

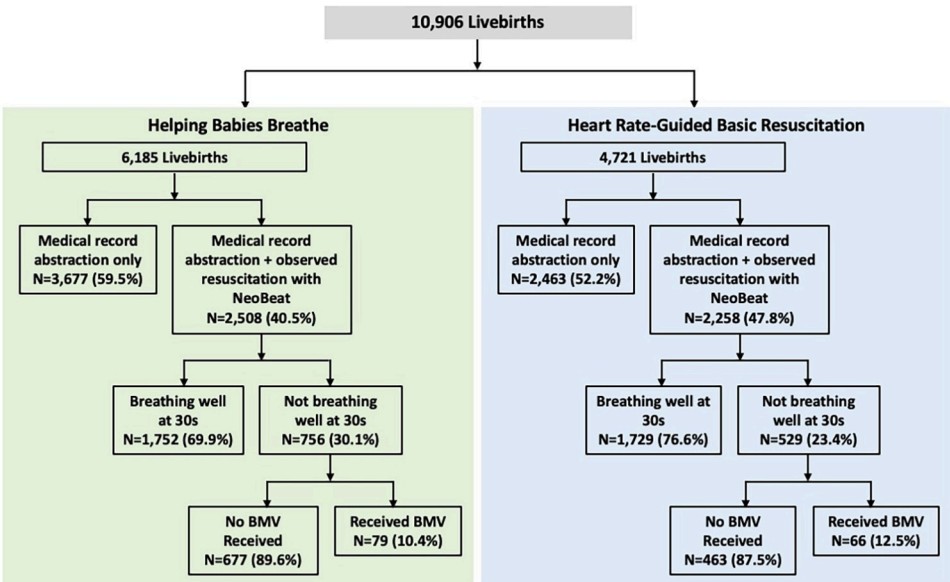

**Fig 2. Flow diagram.** This figure depicts enrollment during HBB (Epoch 2a) versus HR-guided basic resuscitation (Epoch 2b), illustrating the convenience sample of births directly observed with NeoBeat data, the proportion not breathing well and the proportion of those neonates who received BMV.

**Table 2. Demographic characteristics of livebirths.**

| Characteristic n (%) | HBB N = 6,185 | HR-guided basic resuscitation N = 4,721 | p-value of observed differences |
|---|---|---|---|
| **Maternal age** | | | |
| < 20 | 460 (7.5) | 267 (5.7) | 0.0109 |
| 20–35 | 4,821 (78.1) | 3,743 (79.3) | |
| > 35 | 893 (14.5) | 708 (15.0) | |
| Missing | 11 (0.2) | 3 (0.1) | |
| **Parity** | | | |
| 0 | 1,899 (30.8) | 1,427 (30.3) | 0.3477 |
| 1–2 | 2,490 (40.4) | 2,014 (42.8) | |
| 3+ | 1,774 (28.8) | 1,264 (26.9) | |
| Missing | 22 (0.4) | 16 (0.3) | |
| **Mode of delivery** | | | |
| Vaginal without forceps/vacuum | 6,073 (98.3) | 4,690 (99.4) | <0.0001 |
| Vaginal with forceps/vacuum | 6 (0.1) | 1 (0.0) | |
| Cesarean section | 102 (1.7) | 29 (0.6) | |
| Missing | 4 (0.1) | 1 (0.0) | |
| **Birthweight** | | | |
| < 1000g | 3 (0.0) | 5 (0.1) | 0.2699 |
| 1000–1499g | 78 (1.3) | 34 (0.7) | |
| 1500–2499g | 711 (11.5) | 535 (11.3) | |
| ≥ 2500g | 5,388 (87.2) | 4,146 (87.8) | |
| Missing | 5 (0.1) | 1 (0.0) | |
| **Gestational age** | | | |
| < 37 weeks | 1,002 (17.0) | 674 (14.4) | 0.0003 |
| ≥ 37 weeks | 4,905 (83.0) | 4,015 (85.6) | |
| Missing | 278 (4.5) | 32 (0.7) | |
| **Small for gestational age** | | | |
| Yes | 530 (9.0) | 401 (8.6) | 0.4417 |
| No | 5,372 (91.0) | 4,287 (91.4) | |
| Missing | 283 (4.6) | 33 (0.7) | |
| **Multiplicity** | | | |
| Singleton | 5,910 (95.6) | 4,502 (95.4) | 0.6318 |
| Twins | 275 (4.4) | 219 (4.6) | |
| **Newborn sex** | | | |
| Male | 3,111 (50.4) | 2,399 (50.8) | 0.6349 |
| Female | 3,063 (49.6) | 2,319 (49.2) | |
| Missing | 11 (0.2) | 3 (0.1) | |
| **Facility of birth** | | | |
| Facility A | 3,057 (49.4) | 2,711 (57.4) | <0.0001 |
| Facility B | 905 (14.6) | 541 (11.5) | |
| Facility C | 2,223 (35.9) | 1,469 (31.1) | |

data, expert review demonstrated a stillbirth misclassification rate of 5.9% in HR-guided basic resuscitation (n = 1 of 17 stillbirth cases) compared to 33.3% during HBB (n = 6 of 18 stillbirth cases).

There was no difference in the primary outcome of effective breathing at three minutes after birth between groups (aRR 1.08 [95% CI 0.80, 1.45]) nor the secondary outcomes of time

**Table 3. Outcomes of all liveborn infants comparing HBB to HR-guided basic resuscitation.**

| Outcome | HBB<br>N = 6,185<br>n (%) | HR-guided basic resuscitation<br>N = 4,721<br>n (%) | Adjusted RR<br>(95% CI)[a] |
|---|---|---|---|
| **1-minute APGAR score ≤ 5[b]** | 315 (5.1) | 194 (4.1) | 0.90 (0.74, 1.08) |
| **5-minute APGAR score ≤ 5[c]** | 53 (1.3) | 34 (0.8) | 0.66 (0.39, 1.05) |
| **Death before discharge** | 72 (1.2) | 72 (1.5) | 1.59 (1.13, 2.23) |

[a]Adjusted for facility, maternal age, and gestational age

[b]n = 16 (0.3%) missing for HBB, n = 7 (0.2%) missing for HR-guided basic resuscitation

[c]n = 2,222 (35.9%) missing for HBB, n = 460 (9.7%) missing for HR-guided basic resuscitation

to effective breathing (-11.61 seconds [p = 0.11]) and HR≥100 bpm at three minutes after birth (aRR 0.95 [95% CI 0.69, 1.29]; Table 4). The time to HR≥100 bpm significantly decreased by 7.69 seconds during HR-guided basic resuscitation (p = 0.02).

Among newborns receiving BMV, there was no difference in time to effective breathing nor time to HR≥100 bpm (Table 5). The time to BMV decreased by 64.5 seconds from a median of 327 seconds from birth to a median of 283 seconds from birth (p<0.001; Table 5 and Fig 3) There was no difference in the time to improve BMV nor the time to achieve HR>140 bpm after initiating BMV. Providers dried/stimulated newborns 6.8 seconds sooner (p = 0.01); providers practices related to suction (initiation time, total duration, number of episodes) did not change.

## Discussion

In this pre-post interventional trial, we found no difference in the primary outcome of effective breathing at three minutes after birth comparing HR-guided basic resuscitation to HBB. Among newborns not breathing by 30 seconds, the time to HR≥100 bpm decreased by eight seconds during HR-guided basic resuscitation. For cases receiving BMV, providers reduced the time to BMV by over one minute; they also reduced the time to drying/stimulating by six seconds.

**Table 4. Outcomes of observed liveborn newborns with heart rate data from NeoBeat who were not breathing well at 30 seconds after birth comparing HR-guided basic resuscitation to HBB.**

| Outcome | HBB<br>N = 756[a]<br>n (%) | HR-guided basic resuscitation<br>N = 529[b]<br>n (%) | Adjusted RR<br>(95% CI)[c] | Adjusted<br>β (p-value)[d] |
|---|---|---|---|---|
| *Primary outcome* | | | | |
| **Effectively breathing at 3 minutes after birth, n (%)** | 612 (81.0) | 431 (81.5) | 1.08 (0.80, 1.45) | |
| *Secondary outcomes* | | | | |
| **Time to effective breathing, median in seconds (IQR)[e]** | 55 (38, 102) | 49 (37, 83) | | -11.61 (0.11) |
| **HR ≥100bpm at 3 minutes after birth, n (%)** | 612 (83.7) | 417 (82.6) | 0.95 (0.69, 1.29) | |
| **Time to HR ≥ 100bpm, median in seconds (IQR)** | 45 (31, 67) | 35 (26, 51) | | -7.69 (0.02) |

HBB = Helping Babies Breathe; HR = heart rate; RR = relative risk; CI = confidence interval;; bpm = beats per minute

[a]756 non-breathing newborns represent 30.2% of all observed livebirths (n = 2508 total observations)

[b]529 non-breathing newborns represents 24.1% of all observed livebirths (n = 2258 total observations)

[c]Adjusted for facility, maternal age and gestational age

[d]From the generalized linear model, β reflects the absolute value of change with HR-guided resuscitation; adjusted for facility

[e]IQR = interquartile range

**Table 5. Resuscitation practices for observed liveborn newborns not breathing well at 30 seconds after birth who also received BMV comparing HR-guided basic resuscitation to HBB.**

| Characteristic Median in seconds (IQR) | HBB N = 79 | HR-Guided Basic Resuscitation N = 66 | Adjusted β (p-value)[a] |
|---|---|---|---|
| *Physiologic outcomes* | | | |
| **Time to effective breathing** | 249 (134, 517) | 162 (60, 427) | -61.63 (0.16) |
| **Time to HR≥100bpm** | 84 (41, 144) | 79 (39, 164) | 14.92 (0.36) |
| *BMV* | | | |
| **Initiation time from birth** | 327 (248, 422) | 283 (170, 372) | -64.5 (<0.001) |
| **Initiation by 60 seconds after birth, n (%)** | 1 (1.3) | 2 (3.0) | |
| **Total duration of all episodes** | 85 (50, 189) | 64 (28, 147) | -23.89 (0.40) |
| **Time to improve BMV following initiation of BMV[b]** | 38 (26, 70) | 37 (21, 59) | 7.12 (0.57) |
| **HR<100bpm at time of BMV initiation, n (%)** | 36 (50.0) | 44 (73.3) | |
| **Time from initiation of BMV to HR>140bpm** | 65 (28, 160) | 81 (26, 151) | -19.01 (0.48) |
| *Drying/stimulation* | | | |
| **Initiation time from birth** | 11 (7, 23) | 7 (4, 17) | -6.83 (0.01) |
| **Total duration of all episodes** | 532 (372, 761) | 456 (321, 582) | -81.23 (0.12) |
| *Suction* | | | |
| **Initiation time from birth** | 73 (40, 116) | 52 (28, 75) | -20.36 (0.09) |
| **Total duration of all episodes** | 334 (190, 455) | 336 (211, 527) | 19.51 (0.65) |
| **Number of episodes** | 4 (2, 6) | 4 (3, 6) | -0.53 (0.26) |
| *Cord clamping* | | | |
| **Initiation time from birth** | 170 (51, 236) | 141 (63, 192) | -7.46 (0.60) |

[a]From the generalized linear model, β reflects the absolute value of change with HR-guided resuscitation; adjusted for facility, maternal age and gestational age
[b]Denominator n = 57 (72.0%) for HBB, n = 41 (62.0%) for HR-guided basic resuscitation

While providers were more efficient in intervening for infants who were not breathing at birth during heart-rate guided resuscitation, initiation of BMV remained well beyond the recommended Golden Minute at four minutes and 43 seconds after birth. We were troubled to see that HR-guided basic resuscitation was associated with an increase in death before discharge, but do not believe this reflects less effective resuscitative care given the improved time to BMV. We postulate that the reduction in time to BMV was insufficient to reduce mortality in the setting of several minutes of depression following birth. However, it is also possible that newborns are more resilient to time to BMV than anticipated; while time to BMV was significantly associated with death or prolonged hospitalization in one study in Tanzania [4], subsequent studies have demonstrated fetal HR, initial newborn HR and HR response to ventilation to be strong predictors rather than time to BMV [23, 24]. Given the misclassification rate of stillborn infants during HR-guided basic resuscitation (6% compared to 33% during HBB), providers may have become more likely to resuscitate newborns previously presumed stillborn who also had a higher baseline risk of death. The lack of difference in perinatal mortality between HBB and HR-guided basic resuscitation further supports this possibility.

During HR-guided basic resuscitation, depressed newborns received ventilation earlier with a decrease in time to BMV of over one minute. In a previously published analysis of provider practices during HBB implementation in this trial, we reported that excessive drying/stimulation and excessive suctioning both contribute to delays in initiating BMV [18]. We postulate that training in HR-guided basic resuscitation prompted earlier identification of newborns in need of resuscitation based on HR, thereby ultimately decreasing time to BMV.

While we used a cut-off of HR<100 bpm to prompt ventilation in our HR-guided algorithm, more recent normative data reflecting delayed cord clamping suggests that bradycardia

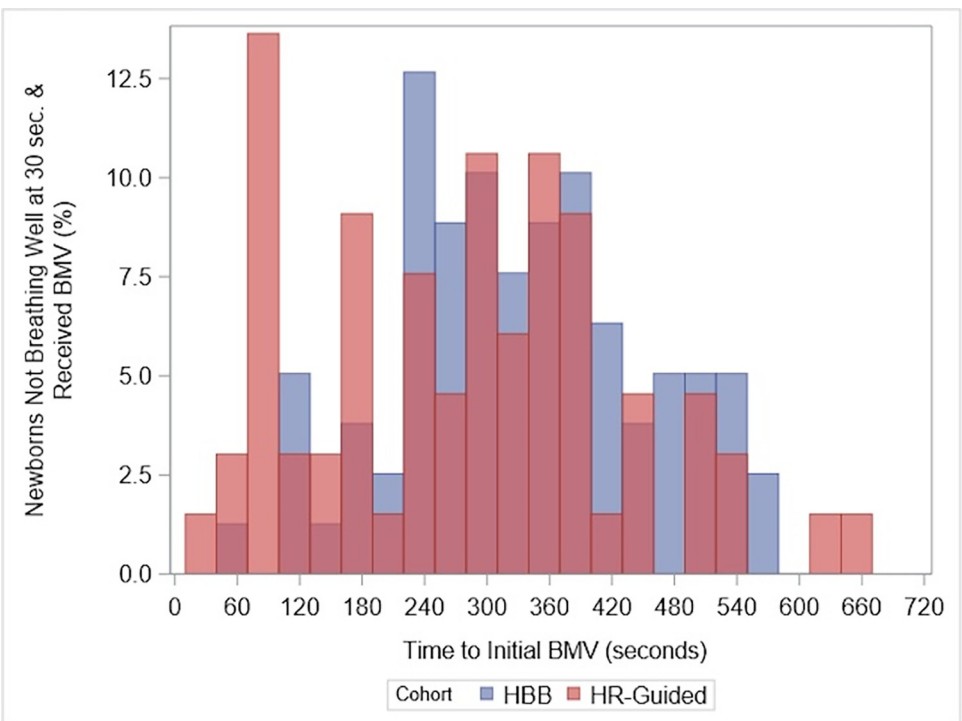

**Fig 3. Time to BMV.** This figure depicts the time between birth and the start of BMV among liveborn newborns not breathing well by 30 seconds after birth, comparing HBB to HR-guided basic resuscitation.

is common in the first minute after birth and mostly self-resolving [25, 26]. Delayed cord clamping, commonly practiced at the facilities in this trial, affords both oxygenation and perfusion via placental circulation while spontaneous respirations are being established. With access to intact placental circulation, the use of dichotomous HR thresholds may over-simplify the identification of neonates who need assisted ventilation.

Resuscitation overall is becoming more complex, with more information providers are expected to assimilate and tasks to manage. As a balancing measure, our data suggests that adding HR did not cause a problem. The changes in care observed with HR-guided basic resuscitation suggest that HBB-trained birth attendants can successfully incorporate HR into their resuscitation practices. Birth attendants may be challenged to evaluate the newborn's respiratory status during resuscitation. Real-time evaluation of breathing during resuscitation is a difficult clinical skill, and a newborn's respiratory status evolves over time during the cardiorespiratory transition after birth. Shifting from a reliance on breathing to an evaluation of breathing supported by objective HR monitoring may aid the provider in accurately and rapidly identifying newborns in need of resuscitation.

Given the pre/post study design, the changes in care observed may be due to alternative explanations rather than HR-guided resuscitation. At the start of HR-guided resuscitation, participants received formal training in resuscitation which included a review of salient steps from HBB. Furthermore, participants engaged in monthly simulation practice of skills throughout the trial. As such, on-going training and simulation practice could have influenced changes in care. Additionally, care following training may naturally improve over time; this phenomenon of time-dependent improvement in care has been previously reported in the First Breath study [27]. Given our relatively short clinical trial and pre-post study design, we

cannot exclude the possibility that on-going training and simulation practice and/or time-dependent improvement influenced the changes in care observed.

Our study is strengthened by the direct observation and detailed collection of the timing of resuscitation care as well as continuous evaluation of the newborn's breathing status along with objective, continuous data on newborn HR. However, we acknowledge limitations of our study. Most importantly, the pre/post study design and lack of randomization does not allow us to distinguish the relative contribution of HR-guided resuscitation versus on-going simulation training to the change in outcomes. Neither participants nor research staff were blinded to the intervention. Several confounding variables may have influenced our results including pre/post differences in baseline maternal, intrapartum and neonatal characteristics, and systematic changes in clinical practice. The evaluation of the newborn's respiratory status was subjectively determined by research staff rather than with a physiologic monitoring device (e.g., impedance monitor). While research staff underwent rigorous training prior to collecting data with the Liveborn app, we did not compare the accuracy of their observations with video recordings of the resuscitations. More than one third of 5-minute Apgar scores were missing during the control phase. Midwives only practiced their resuscitation skills in simulation once monthly throughout the trial; this frequency of practice, while pragmatic, may be insufficient to maximize the benefit of resuscitation training in both HBB and HR-guided basic resuscitation. The relatively short implementation time of 9 months for HBB and 10 months for HR-guided basic resuscitation may not be sufficient to detect improvements in resuscitation care over time [27]. We did not collect data on fetal HR or decision to delivery interval for Cesarean section and assisted vaginal births, thus are unable to comment on how these factors affected the outcome observed [23, 24, 28]. We did not collect data beyond the birth hospitalization; therefore, we do not know the effects of HR-guided basic resuscitation on longer-term outcomes such as neonatal mortality or neurodevelopmental impairment.

## Conclusion

During HR-guided resuscitation, providers were more efficient in responding to non-breathing neonates; however, significant gaps in quality care remain. Our results support the feasibility and potential benefit of integrated HR monitoring in basic resuscitation. Future studies should evaluate the impact of HR-guided basic resuscitation on neonatal morbidity and mortality in a randomized trial.

## Acknowledgments

Jenny Gilbertson was instrumental in producing the HR-guided basic resuscitation algorithm; Ashish Kc facilitated a pilot test of the HR-guided algorithm with midwives in Nepal before we implemented it in this trial.

## Author Contributions

**Conceptualization:** Jackie K. Patterson, Carl Bose, Sara Berkelhamer.

**Data curation:** Pooja Iyer, Tracy Nolen.

**Formal analysis:** Pooja Iyer, Joar Eilevstjønn, Tracy Nolen.

**Funding acquisition:** Jackie K. Patterson, Beena D. Kamath-Rayne, Sara Berkelhamer.

**Investigation:** Daniel Ishoso, Adrien Lokangaka, Eric Mafuta.

**Methodology:** Daniel Ishoso, Adrien Lokangaka.

**Project administration:** Casey Lowman.

**Resources:** Joar Eilevstjønn, Ingunn Haug, Helge Myklebust.

**Software:** Ingunn Haug.

**Supervision:** Adrien Lokangaka, Antoinette Tshefu.

**Visualization:** Joar Eilevstjønn.

**Writing – original draft:** Jackie K. Patterson, Carl Bose, Sara Berkelhamer.

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
