## [Decision Letter · Decision Letter 0]

9 Aug 2024

PONE-D-24-11287Neonatal outcomes and resuscitation practices following the addition of heart rate-guidance to basic resuscitationPLOS ONE

Dear Dr. Patterson,

Thank you for submitting your manuscript to PLOS ONE. After careful consideration, we feel that it has merit but does not fully meet PLOS ONE’s publication criteria as it currently stands. Therefore, we invite you to submit a revised version of the manuscript that addresses the points raised during the review process. The revised version of the manuscript needs to address the methodological limitations of the study and provide a detailed desription of the statistical methods were used for the analysis. 

We look forward to receiving your revised manuscript.

Kind regards,

Dr. Ribka Amsalu

Academic Editor

PLOS ONE

“This work was supported by a Saving Lives at Birth Grand Challenge Award, a Thrasher Early Career Award (PI: Jackie Patterson) and a Laerdal Foundation Award (PI: Sara Berkelhamer).”

“I have read the journal's policy and the authors of this manuscript have the following competing interests: Jackie K. Patterson is the recipient of a Laerdal Global Health gift and in-kind personnel support for an NIH-funded clinical trial in resuscitation in the DRC. Joar Eilevstjønn, Ingunn Haug and Helge Myklebust are all employed by Laerdal Medical, the company that developed NeoBeat. Beena Kamath-Rayne is an employee of the American Academy of Pediatrics, and was an Associate Editor for the Helping Babies Breathe, 2nd Edition. The other authors have no relevant conflicts of interest to disclose.”

6. We note that Figure 1 in your submission contain copyrighted images. All PLOS content is published under the Creative Commons Attribution License (CC BY 4.0), which means that the manuscript, images, and Supporting Information files will be freely available online, and any third party is permitted to access, download, copy, distribute, and use these materials in any way, even commercially, with proper attribution. For more information, see our copyright guidelines: http://journals.plos.org/plosone/s/licenses-and-copyright.

Reviewers' comments:

Reviewer's Responses to Questions

**Comments to the Author**

1. Is the manuscript technically sound, and do the data support the conclusions?

Reviewer #1: Partly

Reviewer #2: No

2. Has the statistical analysis been performed appropriately and rigorously? 

Reviewer #1: I Don't Know

Reviewer #2: No

3. Have the authors made all data underlying the findings in their manuscript fully available?

Reviewer #1: Yes

Reviewer #2: Yes

4. Is the manuscript presented in an intelligible fashion and written in standard English?

Reviewer #1: Yes

Reviewer #2: Yes

5. Review Comments to the Author

Reviewer #1: I congratulate and thank the authors on their important work in improving newborn outcomes. Their paper seeks to evaluate the impact of HR-guided basic resuscitation compared to HBB-alone on neonatal outcomes and resuscitation practices in the DRC. While the study found no significant difference in effective breathing at 3 minutes, it noted a significant reduction in time to bag-mask ventilation by over a minute in the post-intervention phase of the study. I believe the manuscript provides important preliminary data on the potential of HR in guiding newborn resuscitation, but I do have concerns about drawing any conclusions from these data given the complications with study design discussed below.

Title:

1. Clear and accurately reflects the study's content.

Abstract:

2. Concise and effectively summarizes the study's aim, methods, results, and conclusion.

Introduction:

3. Provides a thorough background on the importance of ventilation and HR monitoring during neonatal resuscitation, particularly in low-resource settings. There’s appropriate referencing of the key, relevant articles in the literature. The study’s objective is clearly stated as well as a well-defined primary outcome (effective breathing at 3 minutes after birth).

4. Line 91: The primary outcome measure (effective breathing at 3 minutes after birth) (Line 91) doesn’t seem to match the stated primary objective (evaluating impact on stillbirth misclassification) (Line 101). Wouldn’t it be best for the primary objective to focus on the primary outcome measure? The current primary outcome measure is more aligned with the stated secondary objective (Line 103). Would you like to switch the stated primary and secondary objectives so they better match the primary outcome measure?

Methods:

5. Strengths: Inclusion criteria (≥28 weeks gestation) are clearly defined. The description of the HR-guided resuscitation protocol and training is detailed. The use of observational data and medical records is appropriate, and the study employs rigorous training for data collectors.

6. Concerns: In this particular study, which included interim provider practice (~monthly) and additional training and simulation during the HR training, I have some concern with the pre-post study design being able to distinguish the impact from NeoBeat (discussed below). The lack of randomization may introduce biases. A cluster-randomized trial would strengthen the study's design and is proposed by the authors in their Conclusion. There is no mention of blinding, which could bias the outcomes assessed by the research staff. This could be mentioned in one’s Limitations paragraph.

7. Line 116: I understand it may be described in other publications, but it would likely be useful to readers if the authors provide a brief description of the NeoBeat device (e.g., reusable, hands-free device placed on the chest of the newborn, etc.).

8. Line 117: The authors mention the first phase of the trial included HBB training and that “no guidance on use of HR data to inform resuscitation steps was provided during this control phase.” While I agree that much of the focus of HBB is on newborn breathing rather than HR, the HBB training materials also mention HR evaluation repeatedly — and HR is the focus of the last part of the Action Plan, in particular, beginning on Page 38 of the HBB2 Provider Manual. I would suggest the authors please reword this sentence.

9. Line 132: As the article is written for an international audience, I would suggest the authors provide a reference for NRP and/or cite it as the training program of the AAP.

10. Line 171: Minor comment, but “data” is technically a plural noun so its verb (“was”) should be plural.

11. Line 188: Instead of including “total stillbirths,” shouldn’t the definition of perinatal mortality be “fresh stillbirths + death before discharge”? I don’t believe macerated stillbirths are included in perinatal mortality.

12. Line 196: The authors identify misclassification of a stillborn case includes the presence of a documented HR. However, shouldn’t they be including any sign of life (e.g., a respiratory gasp, any movement, etc.) and not just a documented HR?

Results

13. Strengths: Presented in a structured manner with appropriate use of tables and figures. The statistical methods used, including adjusted relative risk and general linear models, are suitable for the data.

14. Not essential, but additional subgroup analyses (e.g., based on gestational age) could provide deeper insights into which populations benefit most from HR-guided resuscitation.

15. Lines 217-218: Please clarify in the text that this is the *post* OSCE data, as you did in Table 1.

16. Line 219: As above, should be “Data were”

17. Table 2: Could the authors add an additional column that includes p-values comparing HBB to HR-guided basic resuscitation?

18. Table 3: Just a note that there appears to be a fair amount of missing data (e.g., 35.9% of HBB data). This could be briefly mentioned in the Limitations.

Discussion

19. Strengths: Appropriately contextualizes the findings within the broader literature on neonatal resuscitation.

20. Important: Lines 266-268: The authors report that, in addition to decreasing the time to BMV post-intervention, participants also reduced the time to some of the initial HBB steps. These initial resuscitation steps aren’t informed by HR, yet they improved, nonetheless. This suggests to me that these early resuscitation steps and, therefore, at least in part, BMV likely improved because of the interim provider practice and the additional training/practice during the HR training. It’s, therefore, unfortunately tough to say whether any of the improvements in the initial HBB steps or, more importantly, in BMV were attributable to the NeoBeat. I recognize that the authors may be tangentially alluding to this in their discussion of “time-dependent improvement in resuscitation practices” (Lines 313-319). However, I think the issue goes beyond just passive improvement with time; in the post-intervention period (i.e., HR-guidance phase), the providers not only had ~monthly HBB practice sessions but had also received additional HBB training and simulation during the HR training itself. (While I imagine the HR training had a particular focus on the HR portion of the HR-adapted HBB Action Plan, certainly providers were also refreshed to the other portions of the Action Plan, etc. during this HR training.) Therefore, I fear it is quite difficult to ascertain what role NeoBeat played in any improvements in provider skills, correct? Please correct me if I’m missing something. Otherwise, I think the Discussion, Conclusion, and Abstract need to be written in that light, and one couldn’t claim, for example, “HR-guided basic resuscitation resulted in earlier ventilation” (Lines 286-287), “Access to continuous HR data improved time to response…” (Lines 342-344), and other statements in the Discussion, Conclusion, and Abstract.

21. Lines 271-274: Just as a comment, and as the authors also mention, it is troubling that BMV post-intervention was not initiated until an average of 4:43 minutes after birth. In light of the Ersdal et al. statistic we all reference of 16% increased mortality for every 30-second delay in BMV, I am not even sure how much of an impact the one-minute decrease in time to BMV has on improving newborn outcomes after such a long delay. Although it is progress.

22. Lines 272-274: For the reasons cited by the them (in particular, stillborn misclassification and possibly improvement in data collection), I agree with the authors that an increase in death before discharge likely does not reflect less effective resuscitation post-intervention given improvements in other resuscitation parameters.

23. Line 298: Do the authors know whether the rates of delayed cord clamping were roughly similar during the pre vs post phases? DCC may play a role in ‘time to BMV’ if there’s a delay in initially recognizing a non-vigorous newborn and cutting the cord to facilitate BMV. However, if DCC rates were higher in the post-intervention period (which may have occurred with more recent newborn practices), the improvements in time to BMV may be even more noteworthy.

24. Limitations: In addition to the limitation comments above, several confounding variables could influence the results of this study and could be mentioned and/or discussed in the Limitations paragraph, including any pre/post differences in GA, BW, maternal factors, seasonality, concomitant changes in clinical practices, etc.

Conclusion

25. Lines 340-342: As discussed above, I have concern with concluding “Access to continuous HR data improved time to response…”

26. Lines 342-344: I agree with the authors’ proposal for additional studies (e.g., randomized trial), which may address my study-design concerns.

Acknowledgements

27. Minor point, but I believe “implement” should be “implemented”

Reviewer #2: PONE-D-24-11287: statistical review

SUMMARY. This study evaluates the impact of resuscitation training in the Helping-Babies-Breathe protocol (control group) compared to heart-rate-guided resuscitation protocol (intervention group) on effective breathing at three minutes after birth. Secondary outcomes are time to effective breathing, HR≥100 bpm at three minutes after birth, time to HR≥100 bpm, one and five minute APGARs and death before discharge. The statistical analysis seems to rely (not clear from text) on a battery of generalized linear models but statistical methods and materials are poorly presented: see my comments below.

MAJOR POINTS

1. Little is said about the statistical model that has been implemented for the analysis of the primary outcome and such little is also quite obscure. For example, the abstract says "using a method of least squares to fit general linear models". I guess the authors are dealing with generalized (not general) linear models, and these models are fitted by maximum likelihood methods (not least squares). Then lines 198-201 say "We used a method of least squares to fit general linear models using an identity link and binomial distribution assumption to evaluate whether HR-guided basic resuscitation results in an increased probability that infants breathe effectively within three minutes compared to HBB". An identity link function does not make sense under a binomial assumption. Maybe the authors are just misinterpreting statistical terminology, but correct teminology allows to know what exactly has been done, facilitating results reproducibility and interpretation.

2. Although it is not enterly clear (see major point no 1), I guess the authors are fitting a logistic regression model with a canonical logit link for the analysis of the primary outcome. The result are displayed in Table 4 (first row) but they are "adjusted for facility". What does it mean? Is this the "treatment" effect after removing facility-specific differences? In this case, the estimate of beta should be provided along with the facility differences (are they significant?). More generally, it looks like only facility was included as a confounder. What about all the other variables that are summarized in Table 2? Results can be severly biased if these confounders are significatly correlated with the response variable. The coefficient beta should be estimated along with all these confounders.

3. Nothing is said about the statistical models that have been estimated for the analysis of the secondary outcomes. Some of these outcomes are in the form of time up to an event (e.g. time to effective breathing or time to HR≥100 bpm), hence hazard-based regression models are appropriate. Other outcomes are in the form of a binary event (HR≥100 bpm at three minutes after birth or death before discharge) and hence logistic regression is appropriate. The Apgar index is an integer between 0 and 10 and requires a binomial regression or, less rigorously, a linear regression model if the index is approximately normally distributed. Without this information, the estimated treatment effect cannot be interpreted.

4. It seems that the confounders of Table 2 have been ignored in the analysis of the secondary outcomes. Results can be severly biased if these confounders are significatly correlated with the response variable. Relative risks and regression coefficients should be estimated along with all these confounders.

6. PLOS authors have the option to publish the peer review history of their article (what does this mean?). If published, this will include your full peer review and any attached files.

Reviewer #1: **Yes: **Brett D. Nelson, MD, MPH

Reviewer #2: No

---

## [Author Response · Author response to Decision Letter 0]

25 Nov 2024

Response to Reviewer 1 Concerns

Methods

1) The primary outcome measure (effective breathing at 3 minutes after birth) (Line 91) doesn’t seem to match the stated primary objective (evaluating impact on stillbirth misclassification) (Line 101). Wouldn’t it be best for the primary objective to focus on the primary outcome measure? The current primary outcome measure is more aligned with the stated secondary objective (Line 103). Would you like to switch the stated primary and secondary objectives so they better match the primary outcome measure?

Response:

The study we describe in this manuscript is a sub-study of the primary trial; we realize from your comment above that it was confusing to report the objectives per the primary trial. As such, we have simplified our reference to the primary trial and focused on the primary objective of the study described in this manuscript. The methods starting on line 104 of the revised manuscript now reads as follows:

“We conducted a pre-post interventional trial in three urban health facilities in the DRC to evaluate the impact of use of continuous electronic HR monitoring during resuscitation (ClinicalTrials.gov Identifier: NCT03799861). The study design and participants for this trial have been previously described. The primary objective of this study was to evaluate the impact of resuscitation training in HBB (control group) compared to HR-guided basic resuscitation (intervention group) on effective breathing at three minutes after birth.” (Lines 109-110)

2) In this particular study, which included interim provider practice (~monthly) and additional training and simulation during the HR training, I have some concern with the pre-post study design being able to distinguish the impact from NeoBeat (discussed below). The lack of randomization may introduce biases. A cluster-randomized trial would strengthen the study's design and is proposed by the authors in their Conclusion. There is no mention of blinding, which could bias the outcomes assessed by the research staff. This could be mentioned in one’s Limitations paragraph.

Response:

We added these concerns to the limitations paragraph starting on line 512 and reading as follows:

“Most importantly, the pre/post study design and lack of randomization does not allow us to distinguish the relative contribution of HR-guided resuscitation versus on-going simulation training to the change in outcomes. Neither participants nor research staff were blinded to the intervention.” (Lines 512-513)

3) Line 116: I understand it may be described in other publications, but it would likely be useful to readers if the authors provide a brief description of the NeoBeat device (e.g., reusable, hands-free device placed on the chest of the newborn, etc.).

Response:

We have added this content on line 129 of the revised manuscript, which now reads:

“NeoBeat is a re-usable, C-shaped device that measures HR via dry-electrode technology and is easily placed around the thorax of a newborn by a single provider.” (Lines 129-131)

4) Line 117: The authors mention the first phase of the trial included HBB training and that “no guidance on use of HR data to inform resuscitation steps was provided during this control phase.” While I agree that much of the focus of HBB is on newborn breathing rather than HR, the HBB training materials also mention HR evaluation repeatedly — and HR is the focus of the last part of the Action Plan, in particular, beginning on Page 38 of the HBB2 Provider Manual. I would suggest the authors please reword this sentence.

Response:

We have clarified that the guidance provided on HR during HBB training conformed to the standard HBB materials. The methods now read:

“Guidance on use of HR to inform resuscitation steps was provided based on the standard HBB curriculum, which emphasizes evaluation of HR after improving ventilation.” (Lines 131-133) 

5) Line 132: As the article is written for an international audience, I would suggest the authors provide a reference for NRP and/or cite it as the training program of the AAP.

Response:

We added a reference and clarified the program is the AAP’s in the text.

6) Line 171: Minor comment, but “data” is technically a plural noun so its verb (“was”) should be plural.

Response:

Corrected.

7) Line 188: Instead of including “total stillbirths,” shouldn’t the definition of perinatal mortality be “fresh stillbirths + death before discharge”? I don’t believe macerated stillbirths are included in perinatal mortality.

Response:

Perinatal mortality is variably defined in the literature, and sometimes includes all stillbirths but other times is restricted to fresh stillbirths. In the primary trial in which we conducted this sub-study, we elected to use total stillbirths as the primary outcome due to concern for the poor accuracy of clinical exam in determining timing of stillbirth. In keeping with the primary trial, we maintained the same definition of perinatal mortality for this sub-study. We have added this explanation to the methods, which now reads as follows:

“While we did not anticipate HR-guided basic resuscitation to change the outcome of antepartum stillbirths, we elected to include all stillbirths in this definition of perinatal mortality due to concern for the inaccuracy of clinical exam in distinguishing the timing of stillbirth; this decision is also in keeping with the primary trial in which this sub-study was conducted.” (Lines 213-217)

8) Line 196: The authors identify misclassification of a stillborn case includes the presence of a documented HR. However, shouldn’t they be including any sign of life (e.g., a respiratory gasp, any movement, etc.) and not just a documented HR?

Response:

We did not collect data on other signs of life besides breathing. As such, to determine misclassification, we relied solely on objective HR data. If the neonate was classified by the provider as stillborn and there was no HR registered by NeoBeat, we assumed their classification was correct. If the neonate was classified by the provider as stillborn, but NeoBeat registered a HR, we assumed the neonate had been misclassified. To clarify, we deleted the phrase about observational data in this section of the methods. It now reads:

“In post-hoc analysis, we conducted expert review of all stillborn cases for which we had HR data to determine the rate of misclassification based on presence of a documented HR as previously described.” (Lines 222-224)

Results

9) Not essential, but additional subgroup analyses (e.g., based on gestational age) could provide deeper insights into which populations benefit most from HR-guided resuscitation.

Response: 

Given your concerns about teasing out whether changes in practice were due to on-going simulation training versus HR-guided resuscitation, we have elected not to pursue subgroup analyses as we will be unable to draw any firm conclusions from this additional exploration. 

10) Lines 217-218: Please clarify in the text that this is the *post* OSCE data, as you did in Table 1.

Response:

Clarified.

11) Line 219: As above, should be “Data were”

Response:

Corrected.

12) Table 2: Could the authors add an additional column that includes p-values comparing HBB to HR-guided basic resuscitation?

Response:

Added.

13) Table 3: Just a note that there appears to be a fair amount of missing data (e.g., 35.9% of HBB data). This could be briefly mentioned in the Limitations.

Response:

Added as a sentence in the limitations as follows:

“More than one third of 5-minute Apgar scores were missing during the control phase.” (Line 520-521)

Discussion

14) Important: Lines 266-268: The authors report that, in addition to decreasing the time to BMV post-intervention, participants also reduced the time to some of the initial HBB steps. These initial resuscitation steps aren’t informed by HR, yet they improved, nonetheless. This suggests to me that these early resuscitation steps and, therefore, at least in part, BMV likely improved because of the interim provider practice and the additional training/practice during the HR training. It’s, therefore, unfortunately tough to say whether any of the improvements in the initial HBB steps or, more importantly, in BMV were attributable to the NeoBeat. I recognize that the authors may be tangentially alluding to this in their discussion of “time-dependent improvement in resuscitation practices” (Lines 313-319). However, I think the issue goes beyond just passive improvement with time; in the post-intervention period (i.e., HR-guidance phase), the providers not only had ~monthly HBB practice sessions but had also received additional HBB training and simulation during the HR training itself. (While I imagine the HR training had a particular focus on the HR portion of the HR-adapted HBB Action Plan, certainly providers were also refreshed to the other portions of the Action Plan, etc. during this HR training.) Therefore, I fear it is quite difficult to ascertain what role NeoBeat played in any improvements in provider skills, correct? Please correct me if I’m missing something. Otherwise, I think the Discussion, Conclusion, and Abstract need to be written in that light, and one couldn’t claim, for example, “HR-guided basic resuscitation resulted in earlier ventilation” (Lines 286-287), “Access to continuous HR data improved time to response…” (Lines 342-344), and other statements in the Discussion, Conclusion, and Abstract.

Response:

Thanks for this very important comment. We agree that the study design does not allow us to tease out the effect of refresher training and on-going LDHF practice from HR-guided resuscitation. Accordingly, we have tempered our language in the abstract, discussion and conclusion. See changes on lines 47-48, 292-295, 310-311, 339-352, 356-358, and 382-384. 

15) Line 298: Do the authors know whether the rates of delayed cord clamping were roughly similar during the pre vs post phases? DCC may play a role in ‘time to BMV’ if there’s a delay in initially recognizing a non-vigorous newborn and cutting the cord to facilitate BMV. However, if DCC rates were higher in the post-intervention period (which may have occurred with more recent newborn practices), the improvements in time to BMV may be even more noteworthy.

Response:

There was no difference in the time to cord clamping between HBB and HR-guided resuscitation. While the HR-guided resuscitation epoch was not associated with earlier cord clamping, we hypothesize that once neonates were at the warmer, providers were quicker to move on to BMV when they did not respond to suctioning. We have added data on cord clamping to Table 5.

16) Limitations: In addition to the limitation comments above, several confounding variables could influence the results of this study and could be mentioned and/or discussed in the Limitations paragraph, including any pre/post differences in GA, BW, maternal factors, seasonality, concomitant changes in clinical practices, etc.

Response:

We have added a sentence regarding confounding variables to the limitations paragraph as follows:

“Several confounding variables may have influenced our results including pre/post differences in baseline maternal, intrapartum and neonatal characteristics, and systematic changes in clinical practice.” (Lines 513-516)

Conclusion

17) Lines 340-342: As discussed above, I have concern with concluding “Access to continuous HR data improved time to response…”

Response:

We have tempered the first few sentences of the conclusion, which now read as follows:

“During HR-guided resuscitation, providers were more efficient in responding to non-breathing neonates; however, significant gaps in quality care remain. Our results support the feasibility and potential benefit of integrated HR monitoring in basic resuscitation.” (Lines 538-541)

Acknowledgements

18) Minor point, but I believe “implement” should be “implemented”

Response:

Corrected.

Response to Reviewer 2 Concerns (Statistical Review)

1) Little is said about the statistical model that has been implemented for the analysis of the primary outcome and such little is also quite obscure. For example, the abstract says "using a method of least squares to fit general linear models". I guess the authors are dealing with generalized (not general) linear models, and these models are fitted by maximum likelihood methods (not least squares). Then lines 198-201 say "We used a method of least squares to fit general linear models using an identity link and binomial distribution assumption to evaluate whether HR-guided basic resuscitation results in an increased probability that infants breathe effectively within three minutes compared to HBB". An identity link function does not make sense under a binomial assumption. Maybe the authors are just misinterpreting statistical terminology, but correct terminology allows to know what exactly has been done, facilitating results reproducibility and interpretation.

Response:

We were incorrect in stating this was an identity link; we used a canonical logit link for this analysis. For both primary and secondary outcomes, the models used to obtain Adjusted Relative Risks and Beta-estimates are all generalized linear models using maximum likelihood estimation. The models all use a canonical logit link function and now control for (1) epoch (HBB vs. HR-Guided groups), (2) facility, (3) gestational age and (4) maternal age. We have edited the statement in the abstract as follows:

“We evaluated our primary outcome of effective breathing at three minutes after birth among newborns not breathing well at 30 seconds after birth employing generalized linear models using maximum likelihood estimation.” (Lines 34-37)

We have also edited the statement in the methods as follows:

“For both primary and secondary outcomes, we employed generalized linear models using maximum likelihood estimation. The models all used a canonical logit link function and controlled for epoch (HBB vs HR-guided groups), facility, gestational age, and maternal age. All continuous outcomes were evaluated using linear regression and all categorical outcomes were dichotomized and evaluated using logistic regression.” (Lines 210-218)

2) Although it is not entirely clear (see major point no 1), I guess the authors are fitting a logistic regression model with a canonical logit link for the analysis of the primary outcome. The results are displayed in Table 4 (first row) but they are "adjusted for facility". What does it mean? Is this the "treatment" effect after removing facility-specific differences? In this case, the estimate of beta should be provided along with the facility differences (are they significant?). More generally, it looks like only facility was included as a confounder. What about all the other variables that are summarized in Table 2? Results can be severely biased if these confounders are significantly correlated with the response variable. The coefficient beta should be estimated along with all these confounders.

Response:

See response to #1 above. To account for potential variations in resuscitation skills and resource availability across different facilities, we included facility as a control variable in our analysis. This allowed us to isolate the specific impact of HBB and HR-guided training on outcomes while accounting for potential differences in resuscitation skills and facility-level resources that might influence results. 

Per your comment above, we have added gestational age and maternal age as control variables and updated our methods (see lines 210-218) and results accordingly.

3) Nothing is said about the statistical models that have been estimated for the analysis of the secondary outcomes. Some of these outcomes are in the form of time up to an event (e.g. time to effective breathing or time to HR≥100 bpm), hence hazard-based regression models are appropriate. Other outcomes are in the form of a binary event (HR≥100 bpm at three minutes after birth or death before discharge) and hence logistic regression is appropriate. The Apgar index is an integer between 0 and 10 and requires a binomial regression or, less rigorously, a linear regression model if the index is approximately normally distributed. Without this information, the estim

---

## [Decision Letter · Decision Letter 1]

23 Dec 2024

Neonatal outcomes and resuscitation practices following the addition of heart rate-guidance to basic resuscitation

PONE-D-24-11287R1

Dear Dr. Patterson,

We’re pleased to inform you that your manuscript has been judged scientifically suitable for publication and will be formally accepted for publication once it meets all outstanding technical requirements.

Kind regards,

Stefan Grosek, Ph.D., M.D.,

Academic Editor

PLOS ONE

Additional Editor Comments (optional):

Reviewers' comments:

Reviewer's Responses to Questions

**Comments to the Author**

1. If the authors have adequately addressed your comments raised in a previous round of review and you feel that this manuscript is now acceptable for publication, you may indicate that here to bypass the “Comments to the Author” section, enter your conflict of interest statement in the “Confidential to Editor” section, and submit your "Accept" recommendation.

Reviewer #1: All comments have been addressed

Reviewer #2: All comments have been addressed

2. Is the manuscript technically sound, and do the data support the conclusions?

Reviewer #1: Yes

Reviewer #2: (No Response)

3. Has the statistical analysis been performed appropriately and rigorously? 

Reviewer #1: I Don't Know

Reviewer #2: (No Response)

4. Have the authors made all data underlying the findings in their manuscript fully available?

Reviewer #1: Yes

Reviewer #2: (No Response)

5. Is the manuscript presented in an intelligible fashion and written in standard English?

Reviewer #1: Yes

Reviewer #2: (No Response)

6. Review Comments to the Author

Reviewer #1: Thank you to the authors for addressing my (Reviewer 1's) concerns -- particularly my significant concern over confounders being responsible for any changes in outcomes, instead of HR guidance. I defer to our statistician colleague whether all their statistical questions were adequately addressed. Otherwise, I'm comfortable with this revised paper being published.

Reviewer #2: (No Response)

7. PLOS authors have the option to publish the peer review history of their article (what does this mean?). If published, this will include your full peer review and any attached files.

Reviewer #1: **Yes: **Brett D. Nelson, MD, MPH, DTM&H

Reviewer #2: No

---

## [Editor Report · Acceptance letter]

8 Jan 2025

PONE-D-24-11287R1 

PLOS ONE

Dear Dr. Patterson, 

I'm pleased to inform you that your manuscript has been deemed suitable for publication in PLOS ONE. Congratulations! Your manuscript is now being handed over to our production team.

Kind regards, 

on behalf of

Professor Stefan Grosek 

Academic Editor

PLOS ONE